# Functional diversity of rhizosphere soil microbial communities in response to different tillage and crop residue retention in a double-cropping rice field

**Haiming Tang** [ORCID] *, Chao Li, Xiaoping Xiao, Xiaochen Pan, Wenguang Tang, Kaikai Cheng, Lihong Shi, Weiyan Li, Li Wen, Ke Wang

Hunan Soil and Fertilizer Institute, Changsha, China

* tanghaiming66@163.com

## Abstract

Microbial community functional diversity is a sensitive indicator of soil quality, soil management such as tillage and crop residue which can affect the microbial community functional diversity of paddy field. However, there is still limited information about the influence of different tillage and crop residue management on rhizosphere soil microbial community functional diversity in a double-cropping rice (*Oryza sativa* L.) field. Therefore, four tillage treatments were set up in paddy field, tillage treatments were included: conventional tillage with residue incorporation (CT), rotary tillage with residue incorporation (RT), no-tillage with residue retention (NT), and rotary tillage with residue removed as control (RTO). And the effects of CT, RT, NT, and RTO treatments on the average well color development (AWCD), genetic diversity indices and carbon source utilization of rhizosphere soil were studied in the present paper. The results showed that the values of AWCD with CT, RT and NT treatments were higher than that of RTO treatment. It was implied that application of crop residue management resulted in the variation of the carbon utilization efficiency of rhizosphere soil microbial communities. At maturity stages of early and late rice, the Richness indices, Shannon indices and McIntosh indices with CT treatment were significantly higher than that of RTO treatment, and with the order as CT>RT>NT>RTO. Principal component analysis (PCA) results indicated that there were significant differences in carbon substrate utilization patterns among different tillage treatments. Carbohydrates and amino acids were the main carbon resources utilized by rhizosphere soil microbes. Therefore, the combined application of tillage with crop residue management could significantly increase the rhizosphere soil microbial community functional diversity in the double-cropping paddy field of southern China.

**Data Availability Statement:** All relevant data are within the paper.

**Funding:** This study was supported by the National Natural Science Foundation of China (31872851),

the Innovative Research Groups of the Natural Science Foundation of Hunan Province (2019JJ10003), and Hunan Natural Science Foundation of China (2018JJ3305). The funders had no role in study design, data collection and analysis, decision to publish, or preparation of the manuscript.

**Competing interests:** The authors have declared that no competing interests exist.

## Introduction

In recent years, the traditional practice of intensive cultivation involving several deep ploughing and complete removal of crop residue have resulted in low organic matter content, low fertility and high susceptibility to erosion in soils. Some studies indicated that conservation agriculture has been found to increase crop yield, improve water use efficiency, reduce energy inputs, and improve soil fertility [1–2]. Retention of crop residue increases soil organic carbon (C) and nitrogen (N) stocks [3], and the reduction or elimination of tillage reduces soil respiration, resulting in more C in the soil [4,5]. Therefore, soil quality (soil physical and chemical properties, soil microbial biomass content, and so on) has been shown increased under combined application of tillage with crop residue management conditions [6–8]. However, there is still limited information about functional diversity of rhizosphere soil microbial communities in response to different tillage and crop residue retention in a double-cropping rice field.

In all ecosystems, soil microbes play key roles in the decomposition of organic matter, nutrient cycling and altering the availability of nutrients to plants [9], so it has been used as a sensitive indicator to predict soil biological conditions and the effect of agricultural practice in soil ecosystem [10]. The soil microbial community was closely related to the agricultural practices, such as crop residue, organic input, and tillage management, and so on [11]. Understanding the impact of management on the soil microbe community structure and diversity is important for evaluating the effectiveness of a management regime [12]. Frasier et al. [6] found that soil microbial community with crop residue treatments were higher than that of without crop residue input treatment. Yang et al. [13] results found that soil microorganisms to utilize carbon sources with zero-tillage were higher than that of conventional tillage. Wang et al. [14] results found that diversity and stability of soil microbial community were increased with long-term no-tillage and organic input management. Wang et al. [15] results found that soil fungal richness were enhanced with no-tillage and straw mulching management. However, Sirisha et al. [1] results found that soil microbial community were increased under combined application of crop residue with conservation tillage conditions. Other studies found that both soil microorganism metabolic activity, Shannon index and Simpson index were increased with no-tillage and residue retention management [7, 16].

The Yangtze River Plain is one of the most important rice production areas in China [17]. Chinese milk vetch (*Astragalus sinicus* L.) and early rice and late rice (double-cropping rice) (*Oryza sativa* L.) was important crop system in this region. Recently, the manipulation of returning the Chinese milk vetch and rice straw residue as organic fertilizer to the paddy soil was accepted by more and more people because it provides favorable soil environment and nutrient for rice plant growth [18]. Therefore, the amount of application of N and P chemical fertilizer has reduced under adoption of this practice conditions. Meanwhile, rotary tillage and no-tillage with residue retention practices were becoming increasingly popular regionally [8]. However, there is little information about the effects of different tillage and crop residue management on rhizospheric soil microbial communities in the double-cropping rice systems of southern China. We hypothesized that combined application of tillage with crop residue management would affect the rhizosphere soil microbial community function in the double-cropping rice field.

Therefore, the aims of this study were: (1) to analyze the effects on rhizospheric soil microbial activity and function following 4 years of continuous application of conventional tillage, rotary tillage and no-tillage with residue incorporation, and rotary tillage with residue removed, (2) to explore the main carbon resources utilized with different tillage treatments, and (3) to select an appropriate tillage practice for paddy field in a Chinese milk vetch-double cropping rice system.

## Materials and methods

### Study site

In 2015, the experiment was conducted in Ning Xiang County (28°07′ N, 112°18′ E) of Hunan Province, China China, where is the main area of double-cropping rice and there were no endangered or protected species involved. And the experiment under a continental monsoon climate, the annual mean precipitation and evapotranspiration were 1553 and 1354 mm, respectively. The monthly mean temperature was 17.2°C [8]. The predominant soil at the experimental site was a Stagnic Anthrosols and it was developed from the Quaternary red earth (clay loam). The soil texture in the topsoil (0–20 cm) was silt clay loam with 13.85% sand and 56.64% silt. At the beginning of the study, the surface soil characteristics (0–20 cm) were as follows: soil organic carbon (SOC) 22.07 g kg$^{-1}$, total nitrogen (N) 2.14 g kg$^{-1}$, total phosphorous (P) 0.82 g kg$^{-1}$, total potassium (K) 13.21 g kg$^{-1}$, available N 192.20 mg kg$^{-1}$, available P 13.49 mg kg$^{-1}$, and available K 81.91 mg kg$^{-1}$, pH 5.79. There were three crops in a year, Chinese milk vetch (*Astragalus sinicus* L.), early rice and late rice (*Oryza sativa* L.). Chinese milk vetch was sown at the end of October and returned to the paddy field in early April of the following year. Early rice was then transplanted and harvested in the middle of July. Late rice were transplanted in the middle of July and harvested in the end of October.

### Field experiment

The four tillage treatments were initiated in 2015. The systems tested included: conventional tillage with residue incorporation (CT), rotary tillage with residue incorporation (RT), no-tillage with residue retention (NT), rotary tillage with all residue removed as control (RTO). The dimensions of the plots were 56.0 m$_2$ (7 m×8 m), and treatments were laid out in a randomized complete block design with three replications. Chinese milk vetch and rice straw residue were retained for the CT, RT and NT treatments when both the Chinese milk vetch, early and late rice crop residue were retuning to paddy field. The quantity of Chinese milk vetch, early and late rice straw residue added into the paddy soil for the CT, RT and NT treatments were 22500, 2000, and 2000 kg hm$^{-2}$, respectively. The redundant quantity of Chinese milk vetch, early and late rice straw residue removed from the paddy soil for the CT, RT and NT treatments were 29500, 3400, and 4000 kg hm$^{-2}$, respectively, and the quantity of Chinese milk vetch, early and late rice straw residue removed from the paddy soil for the RTO treatment were 52000, 5400, and 6000 kg hm$^{-2}$, respectively. And the carbon content of Chinese milk vetch, early and late rice straw residue were 386.4 g kg$^{-1}$, 395.3 g kg$^{-1}$, and 400.5 g kg$^{-1}$, respectively. No-tillage management was adopted in the NT treatment, and the Chinese milk vetch and rice straw residue were cutted by residue cutting machine and mixed as covering crops on the soil surface before transplanting of rice seedlings. Irrigation water was keep at depth of 2 cm above soil surface with CT, RT and RTO treatments when taken tillage management, then transplanting of rice seedlings. Chinese milk vetch and rice straw residue were cutted by residue cutting machine and incorporated into the soil with tillage management under CT and RT treatments. The CT treatment was tilled once with a moldboard plow to a depth of 15–20 cm and then rotovated twice to a depth of 8–10 cm before transplanting the rice seedlings. The RT and RTO treatments were rotovated four times to a depth of 8–10 cm before transplanting the rice seedlings. The tillage practices with the RTO treatment were similar with that of RT treatment except that Chinese milk vetch and rice straw residue were removed in both early rice and late rice seasons.

 The early and late rice seedlings were manually transplanted to the paddy in April and July, and harvested with a combine in July and October, respectively. The cultivars of early rice

were Xiangzaoxian 45 and the late rice were Xiangwanxian 13 used in the continue three years (2016, 2017 and 2018), respectively. One-month-old seedlings were transplanted at a density of 150,000 plants hm$^{-2}$. The experiment ensured all treatments received the same amount of N, phosphorus pentoxide ($P_2O_5$), potassium oxide ($K_2O$) (the total amount of N, $P_2O_5$, $K_2O$ in chemical fertilizer and that from Chinese milk vetch or rice straw residue) during the early and late rice growing season, respectively. The kinds of fertilizer include urea, ordinary super-phosphate and potassium chloride, respectively. For both early and late rice, the quantity of N were applied at the rate of 150.0 and 180.0 kg hm$^{-2}$ (60% and 40% at basal and tillering stages), 75.0 kg hm$^{-2}$ of $P_2O_5$ as superphosphate, and 120.0 kg hm$^{-2}$ of $K_2O$ as potassium chloride. All the $P_2O_5$ and $K_2O$ fertilizer were applied at tillage before rice transplanting. Paraquat (1, 1′-dimethyl-4, 4′ bipyridiniumion) was applied with 6.0 kg hm$^{-2}$ to control weeds for NT treatment and 1.5 kg hm$^{-2}$ for RT, CT and RTO treatments before the early and late rice transplanting.

## Soil sampling

Soil samples were collected at the maturity stages of early rice and late rice in the middle of late July, and October 2018. Rhizosphere soil was operationally defined as soil adhering to the total roots after gentle shaking. The whole plant with their roots were extracted from soil and after shaking off the loosely adhering soil, the tightly adhering soil (i.e. rhizosphere soil) was care-fully collected. In order to obtain the enough rhizosphere soil for multiplicating, twenty plants were randomly selected from each plot, and these rhizosphere soils were pooled to form one composite sample. Thus, three composite samples of each tillage treatment were collected at sampling time, and a total of 12 composite samples were taken for early rice and late rice matu-rity stages, respectively. The fresh samples were placed immediately in ice box and transported to the laboratory. Large stone or large plant roots were removed by passing the samples through a 2-mm mesh sieve, and the soil samples were then stored at -20˚C until molecular analysis.

## Soil analysis

The functional diversity of rhizosphere soil microorganisms were tested by using Biolog-Eco (Biolog Inc., Hayward, CA, USA) test plates, following a procedure adapted from Garland and Mills [19]. To use the same mass of soil samples for analysis, a certain quantity of fresh soil equivalent to 10.0 g of fresh soil was added to a flask containing 90 ml of sterile water, and then the flask was shaken for 20 min with a speed of 200 r min$^{-1}$ after being sealed. Afterwards, the sterile water was diluted three times by the ten-time dilution method, and 125 μL of the diluted solution was inoculated in each tiny pore of a Biolog-Eco plate. Each treatment had one plate and repeated three times. The inoculated plates were cultured at 25˚C for 12 days, and absorbance was detected by Biolog ELX808 (Hayward, CA, USA) automatic disk reader at 590 nm every 12 h [3]. Data were transformed by using Mierolog 3.4.2 software (BiologInc.). Then the value of average well color development (AWCD) data measured after 120 h of incu-bation were standardized and analyzed by principal component analysis (PCA) [20].

## Statistical analysis

The results of every measured item were presented in mean values and standard error. The data of each treatment means were compared by using one-way analysis of variance (ANOVA) following standard procedures at the 5% probability level. All statistical analyses were calculated by using the SAS 9.3 software package (SAS 9.3) [21]. AWCD and diversity index (Richness, Shannon and McIntosh indices) of different tillage treatments means in this

manuscript were compared by using ANOVA following standard procedures at the $p<0.05$ probability level. Carbon utilization of rhizospheric soil microbial communities with different tillage treatments were determined by PCA, using the PC-ORD software package (Holcomb Research Institute, Butler University) [3].

## Results

### Average well color development

At early rice and late rice maturity stages, AWCD increased with the prolongation of incubation time. There were no changes in AWCD with different tillage treatments at the beginning of 24 h, but increased rapidly within 24–120 h, then increased slowly until the end of the incubation experiment. The results indicated that value of AWCD were rapid increased until 120 h with CT, RT and NT treatments, and the AWCD were ended at 120 h with RTO treatment (Fig 1). There were no significant ($p>0.05$) differences in AWCD between CT, RT, NT and RTO treatments at the beginning 24 h of incubation time. Meanwhile, the results indicated that value of AWCD with CT, RT and NT treatments were higher ($p<0.05$) than that of RTO treatment after 60 h of incubation time.

### Genetic diversity indices of rhizospheric soil microbial communities

Richness, Shannon and McIntosh indices were used to reflect the richness and evenness of rhizospheric soil microbial community species, respectively. At maturity stages of early and late rice, Richness and McIntosh indices with CT treatment were significantly higher ($p = 0.038$, $p = 0.046$) than that of RTO treatment, and with the order as CT>RT>NT>RTO. And the Shannon indices of CT, RT and NT treatments was significantly higher ($p = 0.041$, $p = 0.045$, $p = 0.047$) than that of RTO treatment (Table 1). The results showed that Richness, Shannon and McIntosh indices were increased by application of crop residue management, compared with treatment without crop residue.

### PCA of metabolic function of microbial community in rhizosphere soil

At maturity stages of early and late rice, according to the principle that number of extracted principal components requires the cumulative variance contribution rate to 85% [14], eight principal components were extracted, and the cumulative contribution rates were 84.26% and

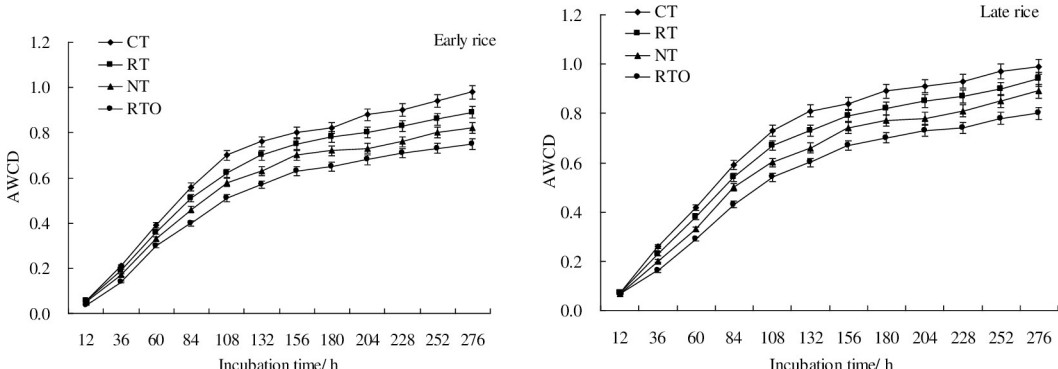

**Fig 1. AWCD changes with incubation progress with different tillage treatments at early rice and later rice maturity stages.** * Abbreviations: CT: conventional tillage with residue incorporation; RT: rotary tillage with residue incorporation; NT: no-tillage with residue retention; RTO: rotary tillage with residue removed. Vertical bars represent the standard error (n = 3). The same as below.

**Table 1. Genetic diversity indices of rhizospheric soil microbial communities with different tillage treatments at early and late rice maturity stages.**

| Rice | Treatments [a] | Items | | |
|------|----------------|-------|---|---|
| | | Richness indices | Shannon indices | McIntosh indices |
| Early rice | CT | 17.15±0.50a | 2.72±0.08a | 6.51±0.18a |
| | RT | 16.86±0.49ab | 2.58±0.08a | 6.26±0.18ab |
| | NT | 16.41±0.47ab | 2.45±0.07a | 5.84±0.16b |
| | RTO | 15.56±0.45b | 2.31±0.06b | 4.93±0.14c |
| Late rice | CT | 16.03±0.46a | 2.62±0.08a | 6.49±0.18a |
| | RT | 15.67±0.45ab | 2.53±0.07a | 6.21±0.17ab |
| | NT | 15.24±0.43ab | 2.44±0.07a | 5.86±0.15b |
| | RTO | 14.53±0.41b | 2.17±0.05b | 4.87±0.13c |

a Treatments: CT: conventional tillage with residue incorporation; RT: rotary tillage with residue incorporation; NT: no-tillage with residue retention; RTO: rotary tillage with residue removed.

Values followed by different small letters within a column are significantly different at $p<0.05$.

86.28%, respectively. At maturity stage of early rice, the variance contribution rates of the first principal component (PC1) and the second principal component (PC2) were 34.37% and 15.58%, and the contribution rates of principal components from 3 to 8 were 8.65%, 7.36%, 5.88%, 4.54%, 4.02% and 3.86%, respectively. At maturity stage of late rice, the variance contribution rates of the PC1 and PC2 were 35.58% and 17.28%, and the contribution rates of principal components from 3 to 8 were 8.12%, 7.05%, 5.24%, 5.09%, 4.27% and 3.65%, respectively. Therefore, only the first two principal components were analyzed in this paper (Fig 2).

The results showed that there were obvious differences on PC axis between different tillage treatments. RTO treatment were distributed in the negative direction of PC1 axis, while CT and RT treatments were distributed in the positive direction of PC1 axis. RTO treatment were distributed in the negative direction of PC2 axis, CT and RT treatments were distributed in the positive direction of PC2 axis, NT treatment were distributes in both positive and negative directions of PC2 axis (Fig 2). Therefore, there were obvious differences in soil microbial communities between different tillage treatments.

The load values of 35 kinds of carbon sources on PC1 and PC2 were further analyzed. The higher of the load values, the more significant the effect of the corresponding carbon sources on the principal components. According to |r|> 0.5, there were 20 kinds of carbon sources contributing to PC1, including 3 kinds of carbohydrates, 9 kinds of amino acids, 3 kinds of carboxylic acids, 1 kind of nucleosides, 2 kinds of phenolic esters, 1 kind of amines and 1 kind of polymers. There were 18 kinds of carbon sources contributing to PC2, including 12 kinds of carbohydrates, 3 kinds of carboxylic acids, 2 kinds of nucleosides and 1 kind of polymer (Table 2). The results showed that carbohydrates and amino acids were the main carbon sources influencing the difference between PC1 and PC2. Amino acid carbon sources have a larger proportion in PC1, while the carbohydrates sources have a larger proportion in PC2. The main carbon sources to distinguish different between tillage treatments were carbohydrates and amino acids.

The first 12 kinds of carbon sources with different tillage treatments were analyzed in the present paper. The results showed that main utilization of rhizosphere soil microorganisms with different tillage treatments was carbohydrates, amino acids and small amount of other substances. But there were fewer similarities of main carbon sources utilization between the different tillage treatments indicating that the utilization of carbon sources by rhizosphere soil microorganisms were changed under different tillage conditions (Table 3).

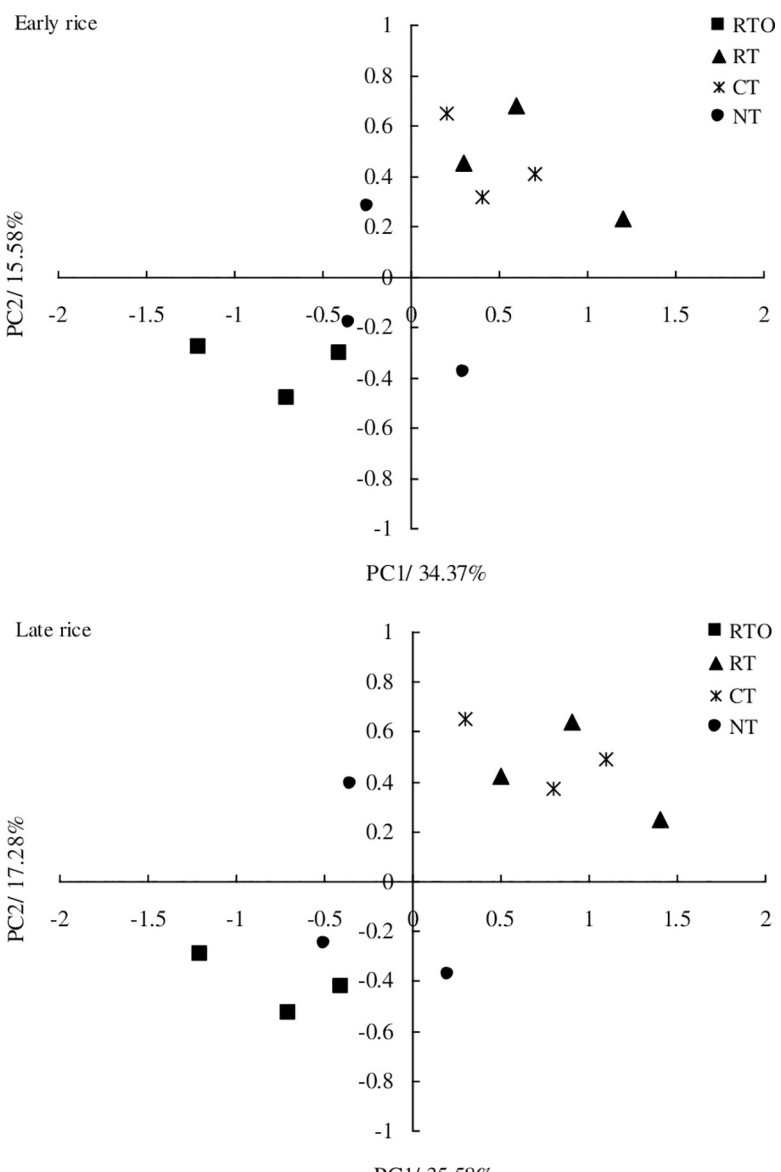

**Fig 2. Principal components analysis for carbon utilization of rhizospheric soil microbial communities with different tillage treatments at early and late rice maturity stages.**

## PCA of carbohydrates, amino acids, carboxylic acids, and nucleosides in rhizosphere soil

L-Serine, D-Glucosaminic acid, and Uridine-5P-monophosphate were the main carbon sources of amino acids, carboxylic acids and nucleosides in CT treatment, respectively. For carbohydrates, there have many kinds of carbon sources, such as D-Mannitol, α-D-Glucose and L-Methyl-D-glucoside, which play a decisive role in CT treatment (Table 2). PCA of carbohydrates, amino acids, carboxylic acids, and nucleosides in rhizosphere soil showed that both the CT and RT treatments were in the carbon source intensive region (Fig 3A, 3B, 3C and 3D). However, the RTO treatment was not in the carbon source intensive region.

**Table 2. Correlation analysis of different carbon source utilization with PC1 and PC2 (the values of |r| < 0.4 were not shown).**

| Carbon source | | PC1 | PC2 |
|---|---|---|---|
| Carbohydrates | D-Mannitol | 0.912 | 0.754 |
| | α-D-Glucose | 0.878 | — |
| | L-Arabinose | — | 0.853 |
| | Glucose-6-phosphate | — | 0.832 |
| | Maltose | — | 0.827 |
| | Lactulose | — | 0.811 |
| | L-Methyl-D-glucoside | 0.854 | 0.805 |
| | D-Trehalose | — | 0.780 |
| | Glucose-1-phosphate | — | 0.791 |
| | D-Fructose | — | 0.785 |
| | D-Raffinose | — | 0.767 |
| | D-Melibiose | — | 0.631 |
| | N-Acetyl-D-glucosamine | -0.462 | 0.625 |
| Amino acids | L-Serine | 0.885 | -0.502 |
| | L-Pyroglutamic acid | 0.881 | — |
| | L-Leucine | 0.875 | — |
| | L-Prolin | 0.870 | — |
| | L-Alanine | 0.865 | — |
| | L-Aspartic acid | 0.851 | -0.467 |
| | L-Asparagine | 0.857 | — |
| | γ-Aminobutyric acid | 0.843 | — |
| | D-Alanine | 0.852 | — |
| Carboxylic acids | D-Glucosaminic acid | 0.902 | — |
| | p-hydroxyphenylacetic acid | 0.872 | — |
| | Quinic acid | 0.886 | — |
| | Sebacic acid | — | 0.786 |
| | Formic acid | — | 0.774 |
| | Malonic acid | — | 0.606 |
| Nucleosides | Uridine-5P-monophosphate | 0.865 | — |
| | Uridine | — | 0.852 |
| | Inosine | — | 0.716 |
| Phenolic esters | Pyruvic acid methylester | 0.853 | -0.416 |
| | Glycerol | 0.803 | — |
| Amine | Putrescine | 0.862 | -0.428 |
| Polymer | Dextrin | 0.807 | 0.576 |

## Discussion

In the present paper, retention of crop residue treatments significantly increased soil microbial activity and functional diversity compared to crop residue removal treatment, indicating that crop residue retention is the key to improve soil microbial community functional diversity in this double cropping paddy soil rather than the elimination of tillage. In this study, the rhizosphere soil microbial community function were changed in combined application of tillage with crop residue soils consistent with our hypothesis that rhizosphere soil microbial community function were increased under combined application of tillage with crop residue conditions, suggesting that soil conditions were enhanced by application of crop residue practices [22]. This conclusion was further reinforced by the similar performance of the crop residue

**Table 3. Main carbon substrates utilized by rhizospheric soil microbial communities with different tillage treatments.**

| Treatments | PC1 | PC2 |
|---|---|---|
| CT | D-L-α-Glycerol phosphate | N-Acetyl-D-galactosamine |
| | D-Sorbitol | i-Erythritol |
| | D-Mannose | L-Serine |
| | Phenylethy lamine | L-Alanyl-glycine |
| | γ-Aminobutyric acid | Glycyl-L-aspartic acid |
| RT | D-Mannose | N-Acetyl-D-glucosamine |
| | Xylitol | N-Acetyl-D-galactosamine) |
| | D-Galactose | Glucose-1-phosphate |
| | L-Pyroglutamic acid | L-Arabinose |
| | Putrescine | Inosine |
| NT | L-Methyl-D-glucoside | β-Methyl-D-glucoside |
| | α-D-Glucose | L-Arabinose |
| | D-Mannitol | Glucose-1-phosphate |
| | D-Galacturonic acid | D-Mannitol |
| | L-Leucine | Sebacic acid |
| RTO | L-Methyl-D-glucoside | L-Arabinose |
| | α-D-Glucose | D-Raffinose |
| | D-Mannitol | Glucose-1-phosphate |
| | L-Prolin | L-Serine |
| | Putrescine | L-Aspartic acid |

retention with tillage treatment, indicating that under these management conditions crop residue exerted significant influence on soil functionality microbial ecology [7, 15].

The value of average well color development (AWCD) is an important indicator of net soil microbial community activity reflecting the utilization ability of single carbon sources [3]. The changing rate of AWCD reflects the ability of soil microorganisms to utilize carbon source. High AWCD value with crop residue retention and tillage treatments indicated that this practice can increase total microbial activity in the rhizosphere soil under a double-cropping rice field of southern China, suggesting that rhizosphere soil microbial growth were increased with good soil environment, moderating moisture and temperature by taking crop residue retention with tillage treatments [1, 13]. The Shannon and McIntosh indices can provide an indication of the soil microbial biocommunity and biodiversity while the Richness indices provide insight into the diversity of microorganisms. In the present study, it was found that all the three indices were increased by taken crop residue retention and tillage management, suggesting that rhizosphere soil microbial communities may be altered with crop residue retention and tillage management, the reason was due to that SOC content and soil ecological environment were increased in crop residue application treatments [8, 14, 23], which provided carbon nutrient and environment for soil microbial multiply [1].

In the present study, under taking the tillage conditions, retaining crop residue in the paddy soil significantly increases most carbon sources and soil microbial diversity, reflecting organic carbon accumulation and the provision of increased substrate supply for soil microorganisms, which was consistent with previous researches [1, 24]. The reason may be that applying conservation tillage and crop residue management increased soil microbial diversity as a result of increase organic carbon from retaining residues. On the other hand, regular addition of SOM may increase background levels of microbial activity, increase nutrient cycling, decrease the concentrations of easily available nutrient sources and increase soil microbial

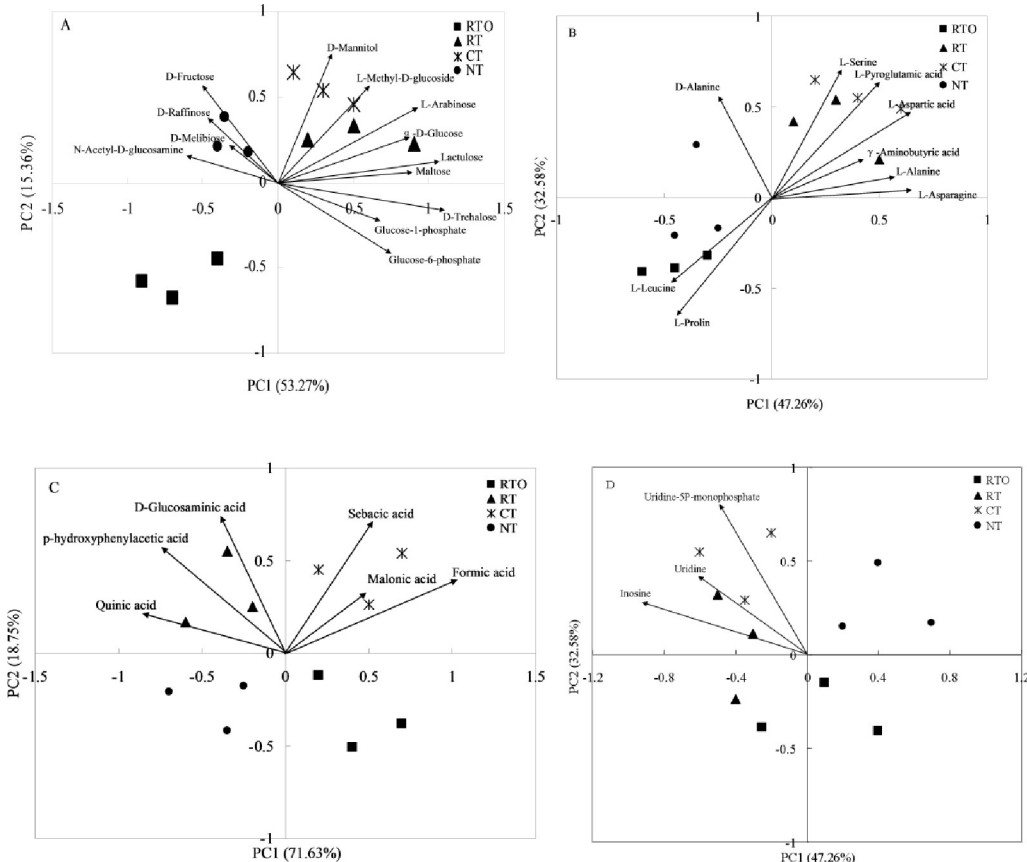

**Fig 3. Principal components analysis of different carbons sources of rhizosphere soil microbial diversity with different tillage treatments (120 h).** * Abbreviations: A indicates carbohydrates, B indicates amino acids, C indicates carboxylic acids, D indicates nucleosides.

diversity [25]. Furthermore, availability of soil carbohydrates is a useful indicator of changes in SOM content [26], and it has been reported that carbohydrates may account up to 75% of the total organic carbon of the soils in some environments and was a key component of the carbon cycle [27]. As such, when the soil microbial community was provided with increased carbohydrates from crop residue retention, activity and functional diversity increases with adaptations for greater carbohydrate utilization, leading to increased carbon cycling and soil fertility.

The positional differences of various samples in the PCA of Biolog-ECO metabolic fingerprints were related to the ability of soil microorganisms to use carbon substrates [19]. In this study, crop residue retention treatments were distributed on the positive axis compared to crop residue removal treatment on the negative axis (Fig 2), with the score coefficients being significantly different, which likely occurred as a result of crop residue retention greatly contributing to rhizosphere soil nutrient by microbial decomposition. Our result supports the hypothesis that application of crop residue would significantly impact rhizosphere soil microbial function consistent with Myers et al. [28], who found that differences in soil microbial functional diversity could be attributed to variations in plant litter quality and substrate inputs to the soil. Meanwhile, the abundant resources and fast nutrient turnover in crop residue retention treatments might contribute to the changes in soil microbial functional diversity [25], and suggested that soil microbial decomposition pathway is relatively more important in the crop residue than in the without of crop residue. In this study, the levels of carbohydrate

and amino acids utilization (PC1 values), including D-Mannitol, α-D-Glucose and L-Methyl-D-glucoside, were higher with crop residue retention treatments. The reason maybe that D-Mannitol constitutes half of the hemicellulose monomer, which is the major constituent of plant cell walls, widely distributed in crop residue of natural ecosystems. Therefore, the results showed that rhizosphere soil microbial functional diversity was all enhanced by retaining crop residue management, indicating that a link between soil nutrient cycling and soil microbial community function exists. However, the mechanism by which these factors interact requires further study.

## Conclusions

The results indicated that ability of rhizosphere soil microorganisms to utilize carbon source were increased by application of crop residue practices regardless of tillage management. It were conducive to maintain the functional diversity of microbial community in rhizosphere soil of paddy field by combined application of conventional tillage, rotary tillage with crop residue management. The species richness and evenness of microbial community in rhizosphere soil were increased with CT and RT treatments. And the carbon substrate utilization patterns were changed with different tillage treatments, the variability of rhizosphere soil microbial community were increased with CT and RT treatments. Carbohydrates and amino acids were the main carbon resources utilized by rhizosphere microbes with different tillage treatments. Therefore, the results indicated that diversity and activity of rhizosphere soil microorganisms in the double-cropping paddy field of southern China were improved by combined application of conventional tillage, rotary tillage with crop residue management.

## Acknowledgments

We acknowledge all the staff members of Hunan Ningxiang County Agricultural and Rural Bureau, and extend special thanks to Yong Li for joining this study.

## Author Contributions

**Funding acquisition:** Haiming Tang.

**Investigation:** Chao Li, Xiaochen Pan, Kaikai Cheng, Lihong Shi, Weiyan Li, Ke Wang.

**Methodology:** Haiming Tang, Xiaoping Xiao, Kaikai Cheng, Li Wen.

**Resources:** Wenguang Tang.

**Software:** Xiaochen Pan, Wenguang Tang, Weiyan Li.

**Supervision:** Kaikai Cheng.

**Writing – original draft:** Haiming Tang.

**Writing – review & editing:** Haiming Tang.

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
