## [Decision Letter · Decision Letter 0]

13 Mar 2020

PONE-D-20-02372

Functional diversity of rhizosphere soil microbial communities in response to different soil tillage and crop residue retention in a double-cropping rice field

PLOS ONE

Dear Dr. Tang,

Thank you for submitting your manuscript to PLOS ONE. After careful consideration, we feel that it has merit but does not fully meet PLOS ONE’s publication criteria as it currently stands. Therefore, we invite you to submit a revised version of the manuscript that addresses the points raised during the review process.

ACADEMIC EDITOR: The study has value but the manuscript has some problems as suggested by the reviewers. The authors should respond to the comments of the reviewers one by one and revise the manuscript accordingly. The revised manuscript would be sent to the reviewers for further reviewing.

We would appreciate receiving your revised manuscript by Apr 27 2020 11:59PM. To enhance the reproducibility of your results, we recommend that if applicable you deposit your laboratory protocols in protocols.io, where a protocol can be assigned its own identifier (DOI) such that it can be cited independently in the future. For instructions see: http://journals.plos.org/plosone/s/submission-guidelines#loc-laboratory-protocols

We look forward to receiving your revised manuscript.

Kind regards,

Jian Liu

Academic Editor

PLOS ONE

Journal Requirements:

- https://www.tandfonline.com/doi/abs/10.1080/09064710.2019.1662082?src=recsys&journalCode=sagb20

-https://www.sciencedirect.com/science/article/pii/S0048969717317564?via%3Dihub

In your revision ensure you quote or rephrase any duplicated text outside the methods section. Further consideration is dependent on these concerns being addressed.

"This study was supported by the National Natural Science Foundation of China (31872851), the Innovative Research Groups of the Natural Science Foundation of Hunan Province (2019JJ10003)."

5. Please include your tables as part of your main manuscript and remove the individual files. Please note that supplementary tables (should remain/ be uploaded) as separate "supporting information" files

Reviewers' comments:

Reviewer's Responses to Questions

**Comments to the Author**

1. Is the manuscript technically sound, and do the data support the conclusions?

Reviewer #1: Partly

Reviewer #2: Partly

Reviewer #3: Yes

Reviewer #4: Partly

2. Has the statistical analysis been performed appropriately and rigorously? 

Reviewer #1: No

Reviewer #2: No

Reviewer #3: Yes

Reviewer #4: No

3. Have the authors made all data underlying the findings in their manuscript fully available?

Reviewer #1: Yes

Reviewer #2: Yes

Reviewer #3: Yes

Reviewer #4: Yes

4. Is the manuscript presented in an intelligible fashion and written in standard English?

Reviewer #1: Yes

Reviewer #2: No

Reviewer #3: Yes

Reviewer #4: No

5. Review Comments to the Author

Reviewer #1: Comments on the manuscript.

Based on field experiments, this manuscript aims to address the effects of soil tillage and residual removal treatments on rhizosphere soil microbial communities. In spite of extensive field works, this study has several intrinsic shortages. First, the experiment design is not full factorial to analyze the effects of soil tillage and crop residue treatments. Second, the literature reviews on the individual effect of soil tillage or residue treatments is not enough. Third, the Biolog data refers to the carbon utilization strategies of the microbial communities. I’m afraid that the indexes (i.e. richness, Shannon, McIntosh indices) in this study based on Biolog data could not be to illustrate the changes in soil microbial community structure. Therefore, I suggest major revisions before the manuscript could be accepted by this journal.

Page 8 line 10, “significantly differences” should be statistically tested.

Page 9 line 11, microbial activity and functional diversity could not be used as the indexes of soil biological fertility.

Page 11 Line 19, please provide evidence from experiments and literatures to support your conclusion that combined application of conventional tillage, rotary tillage with crop residues managements is the most beneficial soil management.

Reviewer #2: In this study, the authors have compared the effects of four treatments of agricultural practice on activity and functional diversity of the soils in paddy fields and the manuscript presents some important findings that combined application of soil tillage and crop residues managements can significantly increase the rhizosphere soil microbial community functional diversity. Generally, the experiment was properly designed and the results are mostly fully presented.

There are several aspects that need to be improved. First, some details of the experiments and the statistical analysis are missing (see below). Second, the language of the manuscript should be carefully improved, as there are many mistakes in the language using.

Introduction

1. P3, Line 1-6. This paragraph seems to be not completed. The authors compare two types of agriculture practices: the traditional practice of intensive cultivation and the conservation agriculture and point out that the conservation practice may be the better choice. But afterwards, the authors do not put forward their scientific question: how to effectively evaluate the soil quality under different agriculture regimes?

Methods:

1. P4, Line 25-29. In this part, what are the residues for Chinese milk vetch if the aboveground is harvested? Maybe it is better to not use ‘harvest’ here.

2. P4, Line 29-30. There is also confusion here. Why were the residues removed for the treatments of CT, RT and NT?

3. P5, Line 17. It would be better to emphasize that the four treatments were continued in the next three years (2016, 2017 and 2018)

4. P6, Line 27-28. the authors state that ‘The most commonly used methods are correlation analysis and regression analysis.’, but no results from any regression analysis are found in the manuscript.

Results:

1.It would be better to present the results of one-way ANOVA of the effects of different tillage treatments on genetic diversity indices of rhizospheric soil microbial communities (F values and p value, etc).

2.P7, Line 5-7. The first two sentences should moved to the Discussion part.

3.P7. Line7-13. It seems that there are no statistics for the AWCD. There are no standard deviation or error for each data point in the Figure 1.The authors also did not present results of any statistical comparison between the four treatments.

Furthermore, the authors state that ‘The changing rate of AWCD reflects the ability of soil microorganisms to utilize carbon source’. It would be better to compare the changing rate of AWCD of the four treatments.

Discussion.

1.P9. Line 15-18. There is a contradictory statement in ‘This conclusion was further reinforced by the similar performance of the crop residues removal treatments with and without tillage, indicating that under these soils and management conditions tillage was significant influence on soil functionality microbial ecology’. If the performance of the crop residues removal treatments with and without tillage were similar, how can the tillage impose significant effects on soil functionality microbial ecology?

Conclusion

1. P11. Line 6-20. The results of the experiment are repeated in this part. Please only summarize all the important findings and their implications.

Language

There are many English grammar mistakes. Please correct them. Some of the examples are:

1. Introduction, P3, Line 1: ’ploughings and complete removal of crop residues has resulted in soils low in organic matter, of poor fertility and very susceptible to erosion.’ is changed to ‘ploughings and complete removal of crop residues have resulted in low organic mater content, low fertility and high susceptibility to erosion in soils ’

2. P3, Line 28: ‘the effect’ changed to ‘the effects’

3. P7 Line 9. ‘There is not change in AWCD with different soil tillage treatments at the beginning of 24 h,’ is changed to ‘There were no changes in AWCD with different soil tillage treatments at the beginning of 24 h,’

4. P8. Line 5-11. ‘were mainly distributes’ is changed to ‘were distributed’.

5. Figure 3 ‘A is indicate carbohydrates, B is indicate amino acids, C is indicate carboxylic acids, D is indicate

nucleosides. ’ is changed to ‘A indicates carbohydrates, B indicates amino acids, C indicates carboxylic acids, D indicates nucleosides. ’

Reviewer #3: Generally, it is a well organized manuscript. But the tense should be consistent. Data analysis part should be clearer, and some content of result should be removed to this part. Delete Results content in conclusion. Add new publications in recent 3 yrs into References. See the reviewed version for details.

Reviewer #4: This paper investigates the interactive effects of soil tillage and crop residues on the microbial community functional diversity. The main results are that crop residues significantly affected the diversity indices and carbon utilization efficiency of rhizospheric soil microbial communities. Also, carbohydrates and amino acids were the main carbon resources utilized by soil microbes. However, as a whole, this paper is in poor written and has big wrong with the statistical analyses. Following are some suggestions:

1. The abstract are in poor organization. For example, the sentences from lns 8-15 in page 2 should be combined and elucidate in clear words. Such as, four treatments (CT, RT, NT, and RTO) were conducted in the field to investigate the average well color development, richness and evenness of the functional diversity, the substrate reaction. Also, the sentence ln 20 recommended to be rewritten as “…..with the order as…..”

2. The introduction should be logically organized by explaining why the soil tillage or crop residues can affect the soil microbes. And then it is concluded that the interactive effects of tillage and crop residues can increase or decrease the soil microbial community diversity. There are many grammatical mistakes in this section. Such as lns 7-8 in page 3, delete the word “in” before nutrient cycling and altering the availability….

3. It is not clear about the experiment design. Especially, how to explore or manipulate the tillage and crop residues addition. Also, it is not clear about the bacterial community analysis using the Biolog ECO microplates. Actually, soil tillage and crop residues are two factors in this experiment, and it should be proper to use two-way ANOVA but not the one-way ANOVA.

4. About the result section, it should state the main findings or results not introduce the meaning of the variables. Such as the sentences lns 5-7, lns 23-25 and the identical sentences in page 7-8.

5. Adherent to the data, the discussion section should clearly and logically explain the results and compared them the other results.

6. The English in the paper needs much improvement. I have made some suggestions but I recommend that a native speaker take a look at the manuscript.

6. PLOS authors have the option to publish the peer review history of their article (what does this mean?). If published, this will include your full peer review and any attached files.

Reviewer #1: No

Reviewer #2: No

Reviewer #3: No

Reviewer #4: No

---

## [Author Response · Author response to Decision Letter 0]

19 Mar 2020

<PLOS ONE >

<PONE-D-20-02372>

Dear Editor, 

Thank you very much for your useful comments and suggestions on our manuscript. Please convey our gratitude to the reviewers who have made useful and detailed suggestions for improvement of the manuscript. As their suggestions we have revised the language and the content with red color in the manuscript. The detail of the changes were listed below in point form:

Reviewer #1: 

1. Page 8 line 10, “significantly differences” should be statistically tested.

√ Page 8 line 10, the “significantly differences” were replaced by “obvious differences”.

2.Page 9 line 11, microbial activity and functional diversity could not be used as the indexes of soil biological fertility.

√ Page 9 line 11, the “biological fertility” were replaced by “microbial community functional diversity”.

3. Page 11 Line 19, please provide evidence from experiments and literatures to support your conclusion that combined application of conventional tillage, rotary tillage with crop residue managements is the most beneficial soil management.

√Page 11 Line 19, these sentences were revised, according to reviewer suggestion.

Reviewer #2: 

Introduction

1. P3, Line 1-6. This paragraph seems to be not completed. The authors compare two types of agriculture practices: the traditional practice of intensive cultivation and the conservation agriculture and point out that the conservation practice may be the better choice. But afterwards, the authors do not put forward their scientific question: how to effectively evaluate the soil quality under different agriculture regimes?

√ P3, Line 1-6, some sentence about the change of soil quality under different agriculture regimes were added, according to reviewer suggestion.

Methods:

1. P4, Line 25-29. In this part, what are the residues for Chinese milk vetch if the aboveground is harvested? Maybe it is better to not use ‘harvest’ here.

√ P4, Line 25-29, some part of Chinese milk vetch residues were returning to paddy filed after Chinese milk vetch harvested, and the quantity of Chinese milk vetch residues returning to paddy filed for the CT, RT and NT treatments were 22500 kg hm-2. Meanwhile, this sentence was revised, according to reviewer suggestion.

2. P4, Line 29-30. There is also confusion here. Why were the residues removed for the treatments of CT, RT and NT?

√ P4, Line 29-30, the total quantity of Chinese milk vetch, early and late rice straw residues for the CT, RT and NT treatments were 52000, 5400, 6000 kg hm-2 both the Chinese milk vetch, early and late rice crops were harvested, respectively. On the one hand, according to the local production habits and the optimum returning amount of Chinese milk vetch, early and late rice straw residues, the quantity of Chinese milk vetch, early and late rice straw residues returning to paddy soil for the CT, RT and NT treatments were 22500, 2000, and 2000 kg hm-2, respectively. On the other hand, the redundant crop residue need remove from the paddy filed. This is, the other quantity of Chinese milk vetch, early and late rice straw residues removed from the paddy soil for the CT, RT and NT treatments were 29500, 3400, and 4000 kg hm-2, respectively.

3. P5, Line 17. It would be better to emphasize that the four treatments were continued in the next three years (2016, 2017 and 2018)

√ P5, Line 17, this sentence were revised, according to reviewer suggestion.

4. P6, Line 27-28. the authors state that ‘The most commonly used methods are correlation analysis and regression analysis.’, but no results from any regression analysis are found in the manuscript.

√ P6, Line 27-28, the sentence of “The most commonly used methods are correlation analysis and regression analysis.” were deleted due to no results from any regression analysis are found in the manuscript, according to reviewer suggestion.

Results:

1.It would be better to present the results of one-way ANOVA of the effects of different tillage treatments on genetic diversity indices of rhizospheric soil microbial communities (F values and p value, etc).

√ The information about the p value of the effects of different tillage treatments on genetic diversity indices of rhizospheric soil microbial communities were added, according to reviewer suggestion.

2.P7, Line 5-7. The first two sentences should moved to the Discussion part.

√ P7, Line 5-7, the first two sentences were moved to the “Discussion ” part, according to reviewer suggestion.

3.P7. Line7-13. It seems that there are no statistics for the AWCD. There are no standard deviation or error for each data point in the Figure 1.The authors also did not present results of any statistical comparison between the four treatments. Furthermore, the authors state that ‘The changing rate of AWCD reflects the ability of soil microorganisms to utilize carbon source’. It would be better to compare the changing rate of AWCD of the four treatments.

√ P7. Line7-13, the standard error for each data point were added in the Figure 1. Meanwhile, the information about statistical compare the changing rate of AWCD with different tillage treatments were added, according to reviewer suggestion.

Discussion.

1.P9. Line 15-18. There is a contradictory statement in ‘This conclusion was further reinforced by the similar performance of the crop residues removal treatments with and without tillage, indicating that under these soils and management conditions tillage was significant influence on soil functionality microbial ecology’. If the performance of the crop residues removal treatments with and without tillage were similar, how can the tillage impose significant effects on soil functionality microbial ecology?

√ P9. Line 15-18, these sentences were revised, according to reviewer suggestion.

Conclusion

1. P11. Line 6-20. The results of the experiment are repeated in this part. Please only summarize all the important findings and their implications.

√P11. Line 6-20, in the “conclusion” section, the structure of this part were revised, some sentences were revised, and the repeated information were deleted, according to reviewer suggestion.

Language

There are many English grammar mistakes. Please correct them. Some of the examples are:

1. Introduction, P3, Line 1: ’ploughings and complete removal of crop residues has resulted in soils low in organic matter, of poor fertility and very susceptible to erosion.’ is changed to ‘ploughings and complete removal of crop residues have resulted in low organic mater content, low fertility and high susceptibility to erosion in soils ’

√Introduction, P3, Line 1, these sentences were revised, according to reviewer suggestion.

2. P3, Line 28: ‘the effect’ changed to ‘the effects’

√ P3, Line 28, the “the effect” were replaced by “the effects”, according to reviewer suggestion.

3. P7 Line 9. ‘There is not change in AWCD with different soil tillage treatments at the beginning of 24 h,’ is changed to ‘There were no changes in AWCD with different soil tillage treatments at the beginning of 24 h,’

√ P7 Line 9, the “There is not change in AWCD with different soil tillage treatments at the beginning of 24 h” were replaced by “There were no changes in AWCD with different soil tillage treatments at the beginning of 24 h”, according to reviewer suggestion.

4. P8. Line 5-11. ‘were mainly distributes’ is changed to ‘were distributed’.

√ P8. Line 5-11, the “were mainly distributes” were replaced by “were distributed”, according to reviewer suggestion.

5. Figure 3 ‘A is indicate carbohydrates, B is indicate amino acids, C is indicate carboxylic acids, D is indicate nucleosides. ’ is changed to ‘A indicates carbohydrates, B indicates amino acids, C indicates carboxylic acids, D indicates nucleosides. ’

√ In Figure 3, the “A is indicate carbohydrates, B is indicate amino acids, C is indicate carboxylic acids, D is indicate nucleosides.” were replaced by “A indicates carbohydrates, B indicates amino acids, C indicates carboxylic acids, D indicates nucleosides.”, according to reviewer suggestion.

Reviewer #3: 

1.The tense of this manuscript should be consistent. 

√ The tense of this manuscript were revised, and keep the tense consistent of this manuscript, according to reviewer suggestion.

2. Data analysis part should be clearer, and some content of result should be removed to this part. 

√ The data analysis part were revised, and some content of result were removed to this part, according to reviewer suggestion.

3. Delete Results content in conclusion. 

√ In the “conclusion” section, some results content were deleted, according to reviewer suggestion.

4. Add new publications in recent 3 yrs into References. 

√ In the “References” section, some references about of new publications in recent 3 year were added, and some old references were replaced by new references, according to reviewer suggestion.

Reviewer #4: 

1. The abstract are in poor organization. For example, the sentences from lns 8-15 in page 2 should be combined and elucidate in clear words. Such as, four treatments (CT, RT, NT, and RTO) were conducted in the field to investigate the average well color development, richness and evenness of the functional diversity, the substrate reaction. Also, the sentence ln 20 recommended to be rewritten as “…..with the order as…..”

√ In the abstract section, P2, Line 8-15, Line 20, these sentences were revised, according to reviewer suggestion.

2. The introduction should be logically organized by explaining why the soil tillage or crop residues can affect the soil microbes. And then it is concluded that the interactive effects of tillage and crop residues can increase or decrease the soil microbial community diversity. There are many grammatical mistakes in this section. Such as lns 7-8 in page 3, delete the word “in” before nutrient cycling and altering the availability….

√ In the introduction section, the introduction were logically organized, some more information about of the soil tillage or crop residues can affect the soil microbes were added, according to reviewer suggestion.

√ Many grammatical mistakes were revised, such as, lns 7-8 in page 3 according to reviewer suggestion.

3. It is not clear about the experiment design. Especially, how to explore or manipulate the tillage and crop residues addition. 

Also, it is not clear about the bacterial community analysis using the Biolog ECO microplates. 

Actually, soil tillage and crop residues are two factors in this experiment, and it should be proper to use two-way ANOVA but not the one-way ANOVA.

√ The manipulate the tillage were introduced in the “Field experiment” section, and the manipulate the crop residues addition were also added in the “Field experiment” section.

√ In the “Soil analysis” section, some more detail information about the bacterial community analysis using the Biolog ECO microplates were added, according to reviewer suggestion.

√ These sentences were revised. In the present study, our proceed on a one-way anova to account for different tillage treatments at the maturity stages of early rice and late rice, respectively. And the CT, RT and NT treatments under the same crop residue conditions, that is, the difference of each investigate items between tillage treatments at the same stage of rice were analysis. 

4. About the result section, it should state the main findings or results not introduce the meaning of the variables. Such as the sentences lns 5-7, lns 23-25 and the identical sentences in page 7-8.

√ In the “result” section, some sentences of lns 5-7, lns 23-25 in page 7-8 were removed to the other places, according to reviewer suggestion.

5. Adherent to the data, the discussion section should clearly and logically explain the results and compared them the other results.

√ In the “Discussion” section, the organization were revised, that is, the discussion section were clearly and logically explain the results of this experiment. Meanwhile, the reason of some results were added, and some references were added to support these conclusion.

6. The English in the paper needs much improvement. 

√ The English grammar of this paper were revised, according to reviewer suggestion.

Journal Requirements:

√ The “key words” section were deleted, the format of this manuscript were revised, such as the format of figures, tables and references were modified, and so on, according to Journal Requirements suggestion.

-https://www.tandfonline.com/doi/abs/10.1080/09064710.2019.1662082?src=recsys&journalCode=sagb20

-https://www.sciencedirect.com/science/article/pii/S0048969717317564?via%3Dihub

In your revision ensure you quote or rephrase any duplicated text outside the methods section. 

√ Two references were added and quoted in the revised manuscript, according to Journal Requirements suggestion. That is, the references about “8. Tang HM, Li C, Xiao XP, Tang WG, Cheng KK, Pan XC, et al. Soil physical and chemical quality as influenced by soil tillage managements under double cropping rice system of southern China. Acta Agr Scand B-S P. 2020; 70: 14–23. 14. Wang Y, Li CY, Tu C, Hoyt GD, Deforest JL, Hu SJ. Long-term no-tillage and organic input management enhanced the diversity and stability of soil microbial community. Sci Total Environ. 2017; 609: 341–347.” were added. 

3. Thank you for stating the following in the Acknowledgments Section of your manuscript: "This study was supported by the National Natural Science Foundation of China (31872851), the Innovative Research Groups of the Natural Science Foundation of Hunan Province (2019JJ10003)."

Please remove any funding-related text from the manuscript and let us know how you would like to update your Funding Statement. 

√ The funding-related text were removed from the manuscript, and the funding-related text were update in our Funding Statement. That is, the information about “This study was supported by the National Natural Science Foundation of China (31872851), the Innovative Research Groups of the Natural Science Foundation of Hunan Province (2019JJ10003), and Hunan Natural Science Foundation of China (2018JJ3305).” were added in our Funding Statement.

√ The ORCID iD were validated in Editorial Manager.

5. Please include your tables as part of your main manuscript and remove the individual files. Please note that supplementary tables (should remain/ be uploaded) as separate "supporting information" files.

√ The tables were added in the revised manuscript, according to Journal Requirements suggestion.

In the other aspect：

1. The “soil tillage” were replaced by “tillage”.

2. The “crop residues” were replaced by “crop residue”.

3. The “paddy fields” were replaced by “paddy field”.

4. The “the” of some places in this manuscript were deleted.

The revised manuscript has been submitted to your journal. Once again, thank you for your help and support during the process of the improvement of the manuscript and we look forward to your positive response.

Yours sincerely,

Hai-ming Tang

---

## [Decision Letter · Decision Letter 1]

13 Apr 2020

PONE-D-20-02372R1

Functional diversity of rhizosphere soil microbial communities in response to different tillage and crop residue retention in a double-cropping rice field

PLOS ONE

Dear Dr. Tang,

Thank you for submitting your manuscript to PLOS ONE. After careful consideration, we feel that it has merit but does not fully meet PLOS ONE’s publication criteria as it currently stands. Therefore, we invite you to submit a revised version of the manuscript that addresses the points raised during the review process.

ACADEMIC EDITOR: The revised version has been improved.  But the manuscript still have some problems as suggested by the reviewers. The authors should respond to the comments of the reviewers one by one and revise the manuscript accordingly. 

We would appreciate receiving your revised manuscript by May 28 2020 11:59PM. To enhance the reproducibility of your results, we recommend that if applicable you deposit your laboratory protocols in protocols.io, where a protocol can be assigned its own identifier (DOI) such that it can be cited independently in the future. For instructions see: http://journals.plos.org/plosone/s/submission-guidelines#loc-laboratory-protocols

We look forward to receiving your revised manuscript.

Kind regards,

Jian Liu

Academic Editor

PLOS ONE

Reviewers' comments:

Reviewer's Responses to Questions

**Comments to the Author**

1. If the authors have adequately addressed your comments raised in a previous round of review and you feel that this manuscript is now acceptable for publication, you may indicate that here to bypass the “Comments to the Author” section, enter your conflict of interest statement in the “Confidential to Editor” section, and submit your "Accept" recommendation.

Reviewer #2: (No Response)

Reviewer #3: All comments have been addressed

Reviewer #4: All comments have been addressed

2. Is the manuscript technically sound, and do the data support the conclusions?

Reviewer #2: Yes

Reviewer #3: Yes

Reviewer #4: Yes

3. Has the statistical analysis been performed appropriately and rigorously? 

Reviewer #2: Yes

Reviewer #3: Yes

Reviewer #4: Yes

4. Have the authors made all data underlying the findings in their manuscript fully available?

Reviewer #2: Yes

Reviewer #3: Yes

Reviewer #4: Yes

5. Is the manuscript presented in an intelligible fashion and written in standard English?

Reviewer #2: No

Reviewer #3: Yes

Reviewer #4: No

6. Review Comments to the Author

Reviewer #2: There are still two many English grammer mistakes in the manuscript. It is suggested that it should be revised by a native Enlgish speaker or English language service.

P12 Line 9： there is no explanation of what are CT, RT, NT, and RTO in the abstract.

P 12 Line 8-11. ‘the average well color development (AWCD), richness and evenness of the functional diversity, the substrate reaction.’ This description gives too many details of quantifying functional diversity of rhizosphere soil microbial communities. Please only show the most important one.

P12 A general scientific question is still missing here in the first paragraph in the Introduction.

P14-P15. It would be better to present the four different treatments by a figure or a table, which can make it more easily understood by readers.

P17. Line 2-6. More details of how the data analysis was performed should be provided.

P17. Line 9-17. P value of the differences should be provided.

P 21, Line 9-11. Revise this sentence: ‘The results provided a reasonable hypothesis that enhanced soil conditions for increased rhizosphere soil microbial community function contribute due to the application of crop residue practices [22]’.

P 21, Line 20-22. Revise this sentence: ‘The reason maybe that crop residue retention with tillage treatments benefits the soil by increasing the soil environment, moderating moisture and temperature variation across seasons and has been reported favorable to microbial growth in rhizosphere soil [1, 13].’

P 21, Line 26-30. Revise this sentence: ‘suggesting that structure of the rhizosphere soil microbial communities may be altered by these two managements, the reason maybe that concentrations of SOC were significantly higher in the crop residue application treatments [14, 23], and soil ecological environment were also increased [8], which provided carbon nutrient and environment for soil microbial multiply [1]’.

Reviewer #3: The authors have fixed most of the problems in last version. So my recommendation is acceptance for publication.

Reviewer #4: This revised paper has been somehow improved. While, there are still some works needed to be done.

1. Some sentences in the introduction are not clear. Such as Page 3 line 1 rewritten the sentence “ to increased”. Page 3 lines 9-10 “ Frasier et al. results showed that….” is recommended to be replaced by “Frasier et al. found that ”. also Page 3 lines 11-12 “Yang et al. results showed that….”, of which what is the meaning of “most carbon sources of microbial ”. some other similar errors such as “ results showed that or results indicated that….” should be reorganized.

2. In the M & M part, some sentences about the description of study site is somehow tedious and not clear. Such as “this study was carried out on private land , the owner were gave permission to ”. By the way, gave references about the climate of the study site. Rewritten the sentences such as page 5 line 29 “ the applied kinds of were including ” as “the fertilized includes urea, ordinary ”. Some sentences about the soil sample is not clear. For example, “plant roots were removed by passing the samples through a 2-mm mesh sieve”, actually, the size of mesh sieve just can remove large stone or large plant roots in the soil.

3. Still, the statistical analysis process is not clear. PCA is a method and is not a kind of parameters or varialbes.

4. In the discussion, some sentences are not easy to understand. Such as Page 11 lines 9-10 “ provide a reasonable hypothesis that for increased ”; Page 11 line 22 “ has been reported favorable ”; also, more words like “the reason maybe…”will be boring to authors.

5. It is suggested that invite native English speaker or academic researcher to polish the language throughout the whole manuscript.

7. PLOS authors have the option to publish the peer review history of their article (what does this mean?). If published, this will include your full peer review and any attached files.

Reviewer #2: No

Reviewer #3: No

Reviewer #4: No

---

## [Author Response · Author response to Decision Letter 1]

16 Apr 2020

<PLOS ONE >

<PONE-D-20-02372>

Dear Editor, 

Thank you very much for your useful comments and suggestions on our manuscript. Please convey our gratitude to the reviewers who have made useful and detailed suggestions for improvement of the manuscript. As their suggestions we have revised the language and the content with red color in the manuscript. The details of these changes were listed below in point form:

Reviewer #2:

1．P 2 Line 9：there is no explanation of what are CT, RT, NT, and RTO in the abstract.

√ In the “Abstract” section, the information about CT, RT, NT, and RTO were added, according to reviewer suggestion.

2．P 2 Line 8-11. ‘the average well color development (AWCD), richness and evenness of the functional diversity, the substrate reaction.’ This description gives too many details of quantifying functional diversity of rhizosphere soil microbial communities. Please only show the most important one.

√ In the “Abstract” section, this sentence were revised, according to reviewer suggestion.

3．P 2 A general scientific question is still missing here in the first paragraph in the Introduction.

√ In the first paragraph of “Abstract” section, a general scientific question were added, according to reviewer suggestion.

4．P 4-5. It would be better to present the four different treatments by a figure or a table, which can make it more easily understood by readers.

√ In the “Field experiment” section, it is difficult to layout a table to present the different information about four different treatments when I try to present the four different treatments by a table, and the relative information about four different treatments were well described in the present status. Therefore, I think that there is need not to present the four different treatments by a a table.

5．P7. Line 2-6. More details of how the data analysis was performed should be provided.

√In P7. Line 2-6, more information about data analysis were added and revised, according to reviewer suggestion.

6．P7. Line 9-17. P value of the differences should be provided.

√In P7. Line 9-17, P value of the differences were added, according to reviewer suggestion.

7．P 11, Line 15-17. Revise this sentence: ‘The results provided a reasonable hypothesis that enhanced soil conditions for increased rhizosphere soil microbial community function contribute due to the application of crop residue practices [22]’.

√In P 11, Line 15-17, this sentence were revised, according to reviewer suggestion.

8．P 11, Line 20-22. Revise this sentence: ‘The reason maybe that crop residue retention with tillage treatments benefits the soil by increasing the soil environment, moderating moisture and temperature variation across seasons and has been reported favorable to microbial growth in rhizosphere soil [1, 13].’

√In P 11, Line 20-22, this sentence were revised, according to reviewer suggestion.

9．P 11, Line 26-30. Revise this sentence: ‘suggesting that structure of the rhizosphere soil microbial communities may be altered by these two managements, the reason maybe that concentrations of SOC were significantly higher in the crop residue application treatments [14, 23], and soil ecological environment were also increased [8], which provided carbon nutrient and environment for soil microbial multiply [1]’.

√In P 11, Line 26-30, these sentences were revised, according to reviewer suggestion.

Reviewer #4:

1. Some sentences in the introduction are not clear. Such as Page 3 line 1 rewritten the sentence “ to increased”. Page 3 lines 9-10 “ Frasier et al. results showed that….” is recommended to be replaced by “Frasier et al. found that ”. also Page 3 lines 11-12 “Yang et al. results showed that….”, of which what is the meaning of “most carbon sources of microbial ”. some other similar errors such as “ results showed that or results indicated that….” should be reorganized.

√In P 3, Line 1, the “ to increased” were replaced by “increased”, according to reviewer suggestion.

√In P 3 lines 9-10, the “results showed that” were replaced by “found that”, according to reviewer suggestion.

√In P 3 lines 11-12, the “indicated” were replaced by “found”, and this sentence were revised, according to reviewer suggestion.

√Other similar errors were also revised, according to reviewer suggestion.

2. In the M & M part, some sentences about the description of study site is somehow tedious and not clear. Such as “this study was carried out on private land , the owner were gave permission to ”. By the way, gave references about the climate of the study site. Rewritten the sentences such as page 5 line 29 “ the applied kinds of were including ” as “the fertilized includes urea, ordinary ”. Some sentences about the soil sample is not clear. For example, “plant roots were removed by passing the samples through a 2-mm mesh sieve”, actually, the size of mesh sieve just can remove large stone or large plant roots in the soil.

√In the M & M part, P4 lines 11-12, this sentence were revised, according to reviewer suggestion.

√In P4 lines 13-15, the reference about the climate of this study site were added, according to reviewer suggestion.

√In P5 line 29, this sentence were revised, according to reviewer suggestion.

√In P6 line 17, this sentence were revised, according to reviewer suggestion.

3. Still, the statistical analysis process is not clear. PCA is a method and is not a kind of parameters or varialbes.

√In the “Statistical analysis” section, this sentence were revised, according to reviewer suggestion.

4. In the discussion, some sentences are not easy to understand. Such as Page 11 lines 9-10 “ provide a reasonable hypothesis that for increased ”; Page 11 line 22 “ has been reported favorable ”; also, more words like “the reason maybe…”will be boring to authors.

√In P11 lines 9-10, this sentence were revised, according to reviewer suggestion.

√In P11 lines 22, these sentences were also revised, according to reviewer suggestion.

5. It is suggested that invite native English speaker or academic researcher to polish the language throughout the whole manuscript.

√ The language of this manuscript were revised by academic researcher, according to reviewer suggestion. Such as the “managements” were replaced by “management”, the “and” were replaced by “with”, and the “the” in some places were deleted, and so on.

The revised manuscript has been submitted to your journal. Once again, thank you for your help and support during the process of the improvement of the manuscript and we look forward to your positive response.

Yours sincerely,

Hai-ming Tang

---

## [Decision Letter · Decision Letter 2]

7 May 2020

PONE-D-20-02372R2

Functional diversity of rhizosphere soil microbial communities in response to different tillage and crop residue retention in a double-cropping rice field

PLOS ONE

Dear Dr. Tang,

Thank you for submitting your manuscript to PLOS ONE. After careful consideration, we feel that it has merit but does not fully meet PLOS ONE’s publication criteria as it currently stands. Therefore, we invite you to submit a revised version of the manuscript that addresses the points raised during the review process.

ACADEMIC EDITOR: The revised version has been improved.  But the manuscript still have some problems as suggested by the reviewer.

We would appreciate receiving your revised manuscript by Jun 21 2020 11:59PM. To enhance the reproducibility of your results, we recommend that if applicable you deposit your laboratory protocols in protocols.io, where a protocol can be assigned its own identifier (DOI) such that it can be cited independently in the future. For instructions see: http://journals.plos.org/plosone/s/submission-guidelines#loc-laboratory-protocols

We look forward to receiving your revised manuscript.

Kind regards,

Jian Liu

Academic Editor

PLOS ONE

Reviewers' comments:

Reviewer's Responses to Questions

**Comments to the Author**

1. If the authors have adequately addressed your comments raised in a previous round of review and you feel that this manuscript is now acceptable for publication, you may indicate that here to bypass the “Comments to the Author” section, enter your conflict of interest statement in the “Confidential to Editor” section, and submit your "Accept" recommendation.

Reviewer #2: All comments have been addressed

Reviewer #4: All comments have been addressed

2. Is the manuscript technically sound, and do the data support the conclusions?

Reviewer #2: Yes

Reviewer #4: Yes

3. Has the statistical analysis been performed appropriately and rigorously? 

Reviewer #2: Yes

Reviewer #4: Yes

4. Have the authors made all data underlying the findings in their manuscript fully available?

Reviewer #2: Yes

Reviewer #4: Yes

5. Is the manuscript presented in an intelligible fashion and written in standard English?

Reviewer #2: Yes

Reviewer #4: No

6. Review Comments to the Author

Reviewer #2: (No Response)

Reviewer #4: This revision has been improved greatly. While, there are still some works needed to be done:

1. In the abstract section, some sentences are not clear, such as Page 2 lines 8~13 remerge the two sentences about the description about the treatment code (CT, RT, NT, RTO). Page 2 13~14 suggested to rewritten the sentence as “…the values of AWCD with application of crop residue were higher that of without crop residues.”

2. In page 3 lines 13-21, the references style in main text is not right, it is suggested to delete the year. Line 20, delete the word “some”. Also, replace “some studies” by “other…”. In lines 27-28, it will be easy to understand to rewritten the sentence as “the manipulation of returning the Chinese milk….to the paddy soil….because it provides….”.

3. In the M& M section, somehow it will be better to rewritten the sentences in page 4 lines 15-18 as “In 2015, the experiment was conducted in Ning Xiang County….China, where is the main area of double-cropping rice and there were no endangered or protected species involved”. Still, there are other mistakes about the sentences, read through this section and polish the language.

4. Although there are improvements about the description of statistical analysis, the description about the PCA is not clear, such as by which software it is performed. It is suggested to reorganized the sentences page 10 lines 10~12.

5. Still, it is suggested the author to read through the whole manuscript and reduce the grammar or language mistakes.

7. PLOS authors have the option to publish the peer review history of their article (what does this mean?). If published, this will include your full peer review and any attached files.

Reviewer #2: No

Reviewer #4: No

---

## [Author Response · Author response to Decision Letter 2]

8 May 2020

<PLOS ONE >

<PONE-D-20-02372>

Dear Editor, 

Thank you very much for your useful comments and suggestions on our manuscript. Please convey our gratitude to the reviewers who have made useful and detailed suggestions for improvement of the manuscript. As their suggestions we have revised the language and the content with red color in the manuscript. The details of these changes were listed below in point form:

Reviewer #4: 

1. In the abstract section, some sentences are not clear, such as Page 2 lines 8~13 remerge the two sentences about the description about the treatment code (CT, RT, NT, RTO). Page 2 13~14 suggested to rewritten the sentence as “…the values of AWCD with application of crop residue were higher that of without crop residues.”

√ In Page 2 lines 8~13, these sentences were revised, according to reviewer suggestion.

√ In Page 2 lines 13~14, this sentences were revised, according to reviewer suggestion.

2. In page 3 lines 13-21, the references style in main text is not right, it is suggested to delete the year. Line 21, delete the word “some”. Also, replace “some studies” by “other…”. In lines 27-28, it will be easy to understand to rewritten the sentence as “the manipulation of returning the Chinese milk….to the paddy soil….because it provides….”.

√ In Page 3 lines 13~21, the year of the references were deleted, according to reviewer suggestion.

√ In Page 3 lines 21, the “some studies ” were replaced by “other studies”, according to reviewer suggestion.

√ In Page 3 lines 27-28, this sentences were revised, according to reviewer suggestion.

3. In the M& M section, somehow it will be better to rewritten the sentences in page 4 lines 15-18 as “In 2015, the experiment was conducted in Ning Xiang County….China, where is the main area of double-cropping rice and there were no endangered or protected species involved”. Still, there are other mistakes about the sentences, read through this section and polish the language.

√ In the M& M section, in page 4 lines 15-18, these sentences were revised, according to reviewer suggestion. And other places were also revised, according to reviewer suggestion.

4. Although there are improvements about the description of statistical analysis, the description about the PCA is not clear, such as by which software it is performed. It is suggested to reorganized the sentences page 7 lines 10~12.

√ In Page 7 lines 10-12, these sentences were revised, according to reviewer suggestion.

5. Still, it is suggested the author to read through the whole manuscript and reduce the grammar or language mistakes.

√ The grammar or language mistakes of this manuscript were revised, according to reviewer suggestion.

The revised manuscript has been submitted to your journal. Once again, thank you for your help and support during the process of the improvement of the manuscript and we look forward to your positive response.

Yours sincerely,

Hai-ming Tang

---

## [Editor Report · Decision Letter 3]

11 May 2020

Functional diversity of rhizosphere soil microbial communities in response to different tillage and crop residue retention in a double-cropping rice field

PONE-D-20-02372R3

Dear Dr. Tang,

We are pleased to inform you that your manuscript has been judged scientifically suitable for publication and will be formally accepted for publication once it complies with all outstanding technical requirements.

With kind regards,

Jian Liu

Academic Editor

PLOS ONE

Additional Editor Comments (optional):

Page 12,line 16: "China China" . Please delete one word.
---

## [Editor Report · Acceptance letter]

13 May 2020

PONE-D-20-02372R3 

Functional diversity of rhizosphere soil microbial communities in response to different tillage and crop residue retention in a double-cropping rice field 

Dear Dr. Tang:

I am pleased to inform you that your manuscript has been deemed suitable for publication in PLOS ONE. Congratulations! Your manuscript is now with our production department. 

With kind regards,

on behalf of

Dr. Jian Liu 

Academic Editor

PLOS ONE